# TOWARDS ROBUST EVALUATION OF PROTEIN GENERATIVE MODELS: A SYSTEMATIC ANALYSIS OF METRICS

## ABSTRACT

The rapid advancement of protein generative models necessitates robust and principled methods for their evaluation and comparison. As new models of increasing complexity continue to emerge, it is crucial to ensure that the metrics used for assessment are well-understood and reliable. In this work, we conduct a systematic investigation of commonly used metrics for evaluating protein generative models, focusing on quality, diversity, and distributional similarity. We examine the behavior of these metrics under various conditions, including synthetic perturbations and real-world generative models. Our analysis explores different design choices, parameters, and underlying representation models, revealing how these factors influence metric performance. We identify several challenges in applying these metrics, such as sample size dependencies, sensitivity to data distribution shifts, and computational efficiency trade-offs. By testing metrics on both synthetic datasets with controlled properties and outputs from state-of-the-art protein generators, we provide insights into each metric's strengths, limitations, and practical applicability. Based on our findings, we offer a set of practical recommendations for researchers to consider when evaluating protein generative models, aiming to contribute to the development of more robust and meaningful evaluation practices in the field of protein design.

## 1 INTRODUCTION

The field of protein generative modeling has witnessed significant progress in recent years, fueled by advancements in machine learning and artificial intelligence Wu et al. (2021); Ovchinnikov & Huang (2021). Various approaches, including language model-based architectures, GANs, VAEs, and diffusion models, have been proposed and successfully applied to generate novel protein sequences Ferruz et al. (2022); Madani et al. (2023); Shin et al. (2021); Repecka et al. (2021); Sevgen et al. (2023); Lin & AlQuraishi (2023); Watson et al. (2023); Alamdari et al. (2023); **?**. Several studies have demonstrated the potential of these models to produce functional proteins that have been validated experimentally in the lab, suggesting that generative models have much to offer in the realm of protein design.

Despite these achievements, the evaluation of protein generative models remains a challenge. Unlike other domains such as image or text generation, where well-established evaluation metrics exist, the protein design field lacks a standardized and comprehensive set of metrics Joshua Southern & Correia (2023). As a result, many studies resort to developing their own ad hoc metrics, leading to inconsistencies and difficulties in comparing results across different models and methods. Moreover, the very question of what constitutes a "good" protein is not trivial, as it involves multiple dimensions such as foldability, structural similarity to natural proteins, and functional relevance.

In this work, we address this gap by defining the desired properties of both sequence protein generative models and the metrics used to evaluate them. We study a set of evaluation metrics for assessing protein quality, diversity, and distributional similarity between generated and natural proteins. We examine the sensitivity of these metrics to different types of data perturbations, explore their interrelationships and underlying assumptions, and evaluate their computational efficiency. Our findings reveal potential pitfalls and weaknesses in current evaluation practices and we provide researchers with practical recommendations to address these issues.

## 2 BACKGROUND AND EVALUATION CRITERIA FOR PROTEIN GENERATIVE MODELS

### 2.1 APPROACHES TO EVALUATING GENERATIVE MODELS

Evaluating generative models presents unique challenges compared to their discriminative counterparts. Unlike discriminative models $P(L|X)$, where we can directly assess performance using labeled data $\{X_i, L_i\}$, generative models $P(X)$ require more nuanced evaluation techniques. These techniques typically fall into two categories, depending on the available data: scenarios where we have only generated samples $\{Y_i\}$, and those where we also have access to the training data $\{X_i\}$.

In the absence of training data, evaluation focuses primarily on the quality and diversity of the generated samples. Quality assessment examines how well the generated samples adhere to desired characteristics, while diversity measures the variability within the generated set. These properties are often evaluated using predefined functions or oracle models that capture domain-specific criteria. When training data is available, we can extend our evaluation to include fidelity and coverage. Fidelity measures the degree to which generated samples resemble real data, while coverage assesses whether the generated samples span the full variability of the real data distribution. Researchers have developed various approaches to quantify these properties, including density and coverage metrics Kynkäänniemi et al. (2019); Naeem et al. (2020), Fréchet Distance Heusel et al. (2017), and Maximum Mean Discrepancy (MMD) Gretton et al. (2012).

To effectively measure the quality of generated protein samples, we must establish a framework for what constitutes a "good" protein. Recent advancements in machine learning for proteins have yielded models capable of both protein folding and inverse folding, allowing us to define a bidirectional mapping between sequence and structure spaces:

$$f : S \to T \quad \text{(folding function)}$$
$$g : T \to S \quad \text{(inverse folding function)}$$

where $S$ represents the space of protein sequences and $T$ the space of 3D structures.
This bidirectional mapping provides a powerful tool for assessing the quality of generated proteins. We propose that a "good" protein should exhibit the following properties:

1) **Structural stability**: For a generated sequence $s$, the predicted structure $f(s)$ should be energetically favorable and stable, as measured by established biophysical metrics. For a generated structure $t$, it should directly exhibit these favorable properties.

2) **Self-consistency**: For a generated protein $x$, which could be either a sequence $s \in S$ or a structure $t \in T$, the composition of the folding and inverse folding functions should approximately recover the original input:
$$\|x - (h \circ k)(x)\| < \epsilon$$
where $(h, k) = (g, f)$ if $x \in S$, and $(h, k) = (f, g)$ if $x \in T$, and $\epsilon$ is a small threshold. This property ensures that the generated proteins are consistent with our understanding of the sequence-structure relationship, regardless of whether the model generates sequences or structures.

These properties ensure that individual generated proteins are physically plausible and consistent with our understanding of the sequence-structure relationship in proteins.

### 2.2 EVALUATING MODEL PERFORMANCE

When evaluating protein generative models, researchers typically examine both the quality of individual generated proteins and the overall performance of the model. While there is no universally agreed-upon approach, several aspects are often considered important:

1) **Fidelity**: The degree to which generated proteins resemble those in the training data, measured through various similarity metrics in both sequence and structure space.

2) **Diversity**: The variability within the set of generated proteins ensures that the model is not simply memorizing the training data.

3) **Novelty**: The model's ability to generate proteins similar to, but not identical to, known proteins.

We can either compute these metrics directly or use distributional similarity metrics, like Frechet distance or MMD, that implicitly account for these properties.

## 2.3 DESIRABLE PROPERTIES OF EVALUATION METRICS

When assessing protein generative models, the choice of evaluation metrics is crucial. Regardless of the specific metrics employed, certain properties are generally desirable. These properties help ensure that our assessments are meaningful, practical, and informative:

1) **Robustness and Sensitivity**: Metrics should strike a balance between robustness to noise and sensitivity to meaningful differences in model performance. They should be resilient to small, random perturbations in the data or model outputs, ensuring that the evaluations are reliable and not overly influenced by the stochastic nature of generative models. Simultaneously, these metrics should remain responsive to significant improvements or differences between models.

2) **Interpretability**: Metrics should provide insights into model performance, allowing researchers to identify specific areas for improvement.

3) **Computational Efficiency**: Evaluation metrics should be computationally tractable, especially when monitoring during model training. The efficiency of a metric depends on both the algorithm itself and its sensitivity to sample size.

These properties are not absolute requirements, but rather guiding principles. The relative importance of each may vary depending on the specific research context and goals. In our subsequent discussion of experiments, we will explore how various metrics align with these desirable properties and their effectiveness in evaluating protein generative models.

## 3 METRICS

Building upon the concepts of protein quality and generative model evaluation, we examine specific metrics used in assessing protein generative models. These metrics fall into three categories: quality metrics, which evaluate individual generated proteins; diversity metrics, which measure variability within the generated set; and distributional similarity metrics, which compare generated and natural protein distributions. Here, we outline the studied metrics; for a detailed description of each metric, refer to Appendix A.

### 3.1 QUALITY METRICS

Quality metrics assess the characteristics of individual generated proteins. We examine four widely used metrics: pLDDT, perplexity, pseudoperplexity, and scPerplexity. These metrics evaluate protein quality from perspectives of structural stability and sequence plausibility.

**The predicted Local Distance Difference Test** (pLDDT) is widely used in protein structure prediction and has become a standard for evaluating the quality of generated proteins Jumper et al. (2021); Watson et al. (2023); Alamdari et al. (2023). AlphaFoldJumper et al. (2021); Ferruz et al. (2022); Nijkamp et al. (2023), ESMFoldLin et al. (2023); Wang et al. (2024); Lin et al. (2024); Hayes et al. (2024) and OmegaFoldWu et al. (2022); Alamdari et al. (2023); Lv et al. (2024) are commonly used for pLDDT prediction. We investigate the impact of model choice and sample size on pLDDT-based quality evaluation.

Adapted from language modeling, **perplexity** (ppl) evaluates sequence quality Madani et al. (2023); Repecka et al. (2021). Perplexity calculations often employ autoregressive transformer protein language models such as ProtGPT2 Ferruz et al. (2022), ProGen2 Nijkamp et al. (2023), and RITA Hesslow et al. (2022). We have found that the perplexity values between these models are highly correlated ($R^2 > 0.92$). Considering this, we use ProGen2-base model for calculating perplexity in this work.

**Pseudoperplexity** (pppl) is an adaptation of perplexity for masked language models Salazar et al. (2019); Lin et al. (2023). The ESM-2 family of bidirectional transformer protein language models is frequently used for pseudoperplexity calculations Lin et al. (2023). We examine the influence of model size (Figure 8) and sample size (Figure 2) on pppl calculations.

**Self-consistency perplexity** (scPerplexity) Alamdari et al. (2023) leverages the sequence-structure relationship: $\text{scPerplexity}(S) = -\log p(S|G(F(S)))$, where $F$ is a folding model and $G$ is an inverse folding model. Lower scPerplexity suggests better alignment between the generated sequence and its predicted structure.

While these metrics provide valuable insights, they each have limitations. pLDDT may underestimate the quality of proteins with intrinsically disordered regions. Perplexity and pseudoperplexity might

be misleading for low-complexity sequences. scPerplexity, while comprehensive, is computationally expensive and depends on the accuracy of both folding and inverse folding models. In the following sections, we empirically evaluate these metrics, assessing their sensitivity, robustness, and correlation. This analysis aims to provide a more nuanced understanding of their strengths and weaknesses in the context of protein generative model evaluation.

## 3.2 DIVERSITY METRICS

Evaluating the diversity of generated protein sequences without reference to training data is non-trivial, yet crucial task in assessing generative model performance. While some studies use average pairwise distances between generated sequences as a diversity measure Watson et al. (2023); Wang et al. (2024); Hayes et al. (2024), this approach often lacks discriminative power. More informative methods employ clustering techniques to analyze sample diversity Lin et al. (2024); Huguet et al. (2024). In this work, we utilize **Cluster Density (CD)** as a diversity metric. CD is defined as the ratio of the number of clusters to the total number of sequences being clustered. We use MMseqs2 Steinegger & Söding (2017) for clustering at 50% and 95% similarity thresholds. The 50% threshold provides insight into the general cluster structure, capturing broader diversity patterns. In contrast, the 95% threshold is sensitive to potential mode collapse scenarios, where a model generates nearly identical sequences.

## 3.3 DISTRIBUTIONAL SIMILARITY METRICS

When evaluating protein generative models with access to training data, we can assess the distributional similarity between generated and real samples. This comparison provides insights into two crucial aspects of model performance: fidelity and diversity Naeem et al. (2020). Fidelity measures the extent to which generated samples accurately represent the characteristics of the real data distribution, typically quantified by the proportion of generated samples that closely resemble real samples. Diversity, on the other hand, assesses the model's ability to capture the full range of variation in the real data, often measured by the proportion of real data samples with close analogs in the generated set.

Two main approaches to quantifying distributional similarity are direct measurement of fidelity and diversity and compound metrics that implicitly account for both. Both approaches typically operate on distributions of protein vector representations, either sequence-based (using protein language models) or structure-based (employing 3D-protein encoders).

Direct measurement approaches include **Improved Precision and Recall** (IPR) and **Density and Coverage** (D&C). IPR, introduced by Kynkäänniemi et al. (2019), operates in a high-dimensional feature space, quantifying both the quality and diversity of generated samples. It defines precision as the proportion of generated samples that closely resemble real samples, and recall as the proportion of real samples that have close analogues in the generated set. Complementing IPR, the D&C metric, proposed by Naeem et al. (2020), offers a more intuitive interpretation of similar concepts. Density measures the proportion of generated samples that fall within the manifold of real data, while coverage assesses the proportion of the real data manifold that is represented by the generated samples. Together, these metrics offer a comprehensive evaluation of a generative model's output, balancing the critical aspects of sample realism and distributional coverage.

Compound metrics implicitly account for both fidelity and diversity, and are more widely used in practice. These include Fréchet distance, Maximum Mean Discrepancy (MMD), and Earth Mover's Distance (EMD). **The Fréchet distance** quantifies dissimilarity between two multivariate Gaussian distributions $d(X_1, X_2)^2 = ||\mu_1 - \mu_2||^2 + \text{Tr}(\Sigma_1 + \Sigma_2 - 2\sqrt{\Sigma_1 \Sigma_2})$, for samples $X_1 \sim \mathcal{N}(\mu_1, \Sigma_1)$ and $X_2 \sim \mathcal{N}(\mu_2, \Sigma_2)$. The Fréchet Inception Distance (FID) Heusel et al. (2017), a variant of this metric, is well-established in image generation tasks. In the context of protein modeling, a similar approach using the ProtT5 Alamdari et al. (2023) or ESM-1v Darmawan et al. (2023) encoder has been applied to protein sequence data.

**Maximum Mean Discrepancy (MMD)** Gretton et al. (2012) measures the distance between two distributions in a reproducing kernel Hilbert space. For two samples $X = x_1, ..., x_n$ and $Y = y_1, ..., y_n$ and a kernel $k$, the empirical estimate of MMD is:

$$MMD_k^2(X, Y) = \frac{1}{n^2} \sum_{i=1}^{n} \sum_{j=1}^{n} (k(xi, xj) + k(yi, yj) - 2k(xi, yj))$$

In our study, we employ the radial basis function (RBF) kernel for MMD calculations. Recently, the use of MMD with a ProteinMPNN encoder was proposed for evaluating 3D protein structures Joshua Southern & Correia (2023).

**Earth Mover's Distance (EMD)** measures the minimum cost of transforming one distribution into another, providing insights into diversity and proximity to the dataset.

These metrics operate on protein vector representations derived from sequence-based or structure-based encoders. We systematically investigate their robustness, reliability, and practical applications in evaluating protein generative models.

## 4 EXPERIMENTS

Our experimental framework aims to provide a comprehensive evaluation of metrics used in assessing protein generative models. We focus on three key aspects: quality metrics, diversity metrics, and distributional similarity metrics. Through a series of controlled experiments and analyses, we investigate these metrics' behavior, sensitivity, and practical applicability under various conditions relevant to protein generation tasks.

### 4.1 EXPERIMENTAL SETUP

To evaluate the behavior and sensitivity of the studied metrics, we employ both synthetic datasets that provide controlled experimental conditions and real-world generated data. Our experimental setting consists of two complementary approaches with synthetic data designed to controllably assess the metrics of quality and diversity.

The first set of experiments focuses on simulating the training progress of generative models. Using the SwissProt dataset, a manually curated collection of high-quality protein sequences, as our reference, we introduce controlled perturbations by randomly substituting amino acids while preserving the overall amino acid distribution. Noise levels range from 0% to 30% in 5% increments. This approach allows us to isolate the impact of sequence quality on our metrics and assess their sensitivity to varying degrees of model undertraining.

Our second experimental setup evaluates metrics for diversity and distributional similarity. We construct a dataset of sequences from five distinct protein families chosen to naturally form well-defined clusters. This design uses the fact that proteins within a family are more closely related to each other than to members of other families.

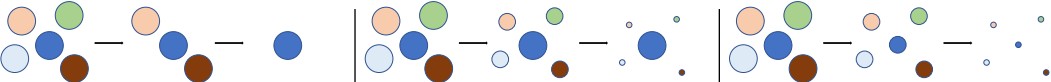

Figure 1: Cluster corruption experiments: Cluster Elimination (left), Cluster Imbalance (middle), and Intra-cluster Diversity Reduction (right).

We introduce three variations to this clustered dataset (Figure 1):
- Cluster Elimination: We sequentially remove entire clusters, simulating mode collapse scenarios where a generative model focuses on an increasingly narrow subset of the protein space.
- Cluster Imbalance: We progressively reduce the presence of four clusters while maintaining one at full size, simulating scenarios where a generative model might overrepresent certain protein families.
- Intra-cluster Diversity Reduction: We gradually replace unique sequences within each cluster with duplicates, maintaining the overall cluster structure while reducing local diversity. This mimics situations where a model produces high-quality but limited-variety sequences within protein families.

These controlled experiments enable us to assess the sensitivity and reliability of our metrics across various scenarios relevant to protein generative modeling, providing insights into their practical application and interpretation.

### 4.2 QUALITY METRICS ANALYSIS

We analyze four widely used quality metrics: predicted Local Distance Difference Test (pLDDT), perplexity (ppl), pseudoperplexity (pppl), and self-consistency perplexity (scPerplexity). Our analysis

focuses on their sensitivity to sample size, their correlation with each other, and the impact of different underlying models on their performance.

### 4.2.1 CORRELATION BETWEEN QUALITY METRICS

To assess potential redundancy among the four quality metrics under consideration (pLDDT, perplexity, pseudoperplexity, and scPerplexity), we conduct a correlation analysis. Our findings reveal varying degrees of interdependence. Perplexity (ppl) and pseudoperplexity (pppl) exhibit a strong correlation ($R^2 > 0.94$), suggesting redundancy in their information content. In contrast, moderate correlations ($R^2 = 0.52$) are observed between pLDDT and ppl, as well as between scPerplexity and ppl. These moderate correlations indicate that while these metrics capture some similar aspects of protein quality, they also provide distinct information, potentially offering complementary insights when used in combination.

The computational demands of pseudoperplexity, requiring masked language model predictions for each sequence position, suggest that standard perplexity may be more practical for most scenarios. Our investigation of pseudoperplexity across ESM-2 model sizes (8M to 3B parameters) reveals high correlations between pppl predictions, despite significant differences in model complexity (Figure 8). This finding has implications for practical applications, as smaller, computationally efficient models may suffice for these calculations.

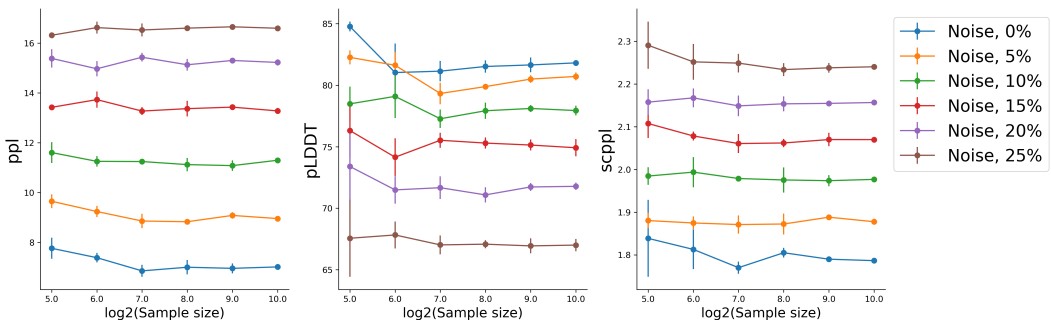

Figure 2: The dependence of quality metrics performance from the sample size.

### 4.2.2 SAMPLE SIZE SENSITIVITY

We analyze the impact of sample size on pLDDT, perplexity, and scPerplexity across various levels of sequence perturbation (0% to 25%) and sample sizes ($2^5$ to $2^{10}$). Figure 2 illustrates our findings. Perplexity, calculated using the ProGen2-base model, demonstrates remarkable consistency across all sample sizes, suggesting its reliability even with limited data. In contrast, pLDDT, calculated using ESMFold, shows greater variability, particularly at smaller sample sizes, stabilizing as the sample size approaches $2^8$ (256) and beyond. ScPerplexity, calculated using ESMFold and ProteinMPNN, exhibits the highest sensitivity to sample size, especially for highly perturbed sequences (15-25% perturbation), with estimates stabilizing around $2^9$ (512) samples. Notably, the relative ordering of perturbation levels remains consistent across sample sizes for all metrics, indicating their ability to differentiate sequence quality in a sensible sample size range. Based on these observations, we recommend a minimum sample size of $2^9$ (512) for robust evaluations, particularly when utilizing scPerplexity or analyzing sequences of unknown quality. This sample size ensures reliable estimates across all examined metrics and perturbation levels.

### 4.2.3 IMPACT OF STRUCTURE PREDICTION MODELS ON PLDDT

We investigate the correlation between pLDDT values produced by AlphaFold, ESMFold, and OmegaFold to assess their interchangeability in quality evaluation (Figure 9). Our analysis focuses on pLDDT values ranging from 50 to 100, as this range is most relevant for assessing protein quality. The results show a strong correlation between the models, with correlation coefficients of 0.857 for OmegaFold and 0.783 for ESMFold when compared to AlphaFold predictions. This high level of agreement suggests that these models can indeed be used interchangeably for quality assessment tasks. However, it's worth noting that ESMFold outperforms OmegaFold in terms of computational efficiency Chen et al. (2024) This efficiency advantage makes ESMFold a particularly attractive option for large-scale evaluations or real-time quality assessment during model training.

### 4.3 DIVERSITY METRICS ANALYSIS

To evaluate the diversity of generated protein sequences without reference to training data, we utilize Cluster Density (CD) as our primary diversity metric. CD is defined as the ratio of the number of clusters to the total number of sequences being clustered. We employ MMseqs2 for sequence clustering, applying two similarity thresholds: 50% and 95%. This dual-threshold approach offers a comprehensive perspective on sequence diversity. Clustering with a high threshold (95%) is used to collapse very close sequences and show the duplication level of the generation. Middle threshold (50%) reveals the structure of generated sample

To assess Cluster Density (CD) sensitivity to decreasing diversity, we conduct an Intra-cluster Diversity Reduction experiment. Figure 3 illustrates CD behavior in this scenario for 50% and 95% similarity thresholds.

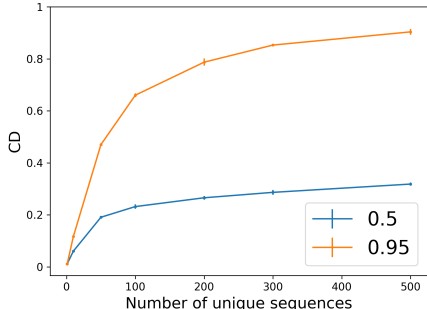

Figure 3: Cluster Density (CD) as a function of unique sequences for 50% and 95% similarity thresholds, demonstrating threshold-dependent sensitivity and saturation effects.

### 4.4 DISTRIBUTIONAL SIMILARITY METRICS ANALYSIS

We evaluate five distributional similarity metrics: Improved Precision and Recall (IPR), Density and Coverage (D&C), Maximum Mean Discrepancy (MMD), Fréchet Distance (FD), and Earth Mover's Distance (EMD). These metrics compare generated protein samples' distribution to real protein samples in a high-dimensional feature space.

#### 4.4.1 PROTEIN REPRESENTATION MODELS

We examine the performance of these metrics across four protein language models of varying sizes: ESM-2 8M, 650M, 3B, and ProtT5. This range allows us to assess the impact of model size and architecture on the behavior of distributional similarity metrics. The behavior of distributional distance metrics, FD, MMD, and EMD, exhibits consistency across different model sizes (Figures 13,14,15,16). This consistency suggests that the choice of embedding model is not critical, considering three orders of magnitude in parameter counts. On the other hand, the fidelity/diversity metrics show erratic behavior in all representations (Figures 17, 18). This suggests that IPR and D&C metrics require extra caution.

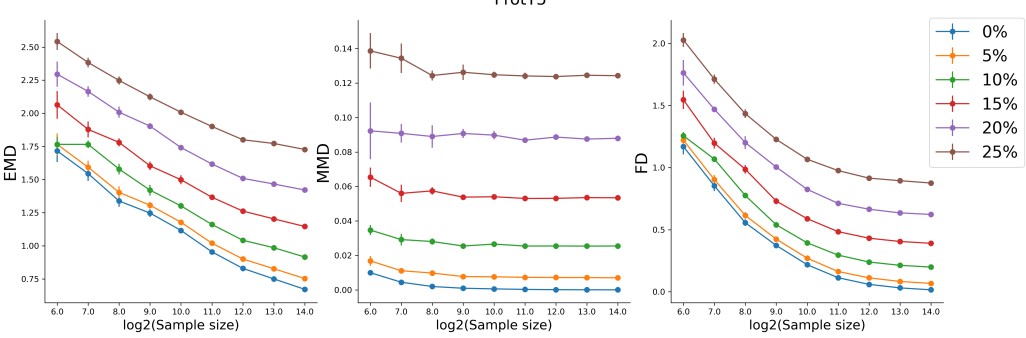

Figure 4: The distributional distances between the original and perturbed data as a function of sample size.

#### 4.4.2 SAMPLE SIZE SENSITIVITY

We systematically evaluate the metrics' behavior across different sample sizes ($2^5$ to $2^{14}$) and perturbation levels (0% to 25%). Our results, depicted in Figures 16, 13, 14, and 15, reveal several key trends. Across all models and metrics, we observe stratification of perturbation levels, with higher perturbation resulting in larger distributional distances. This separation is maintained across sample sizes, indicating that these metrics can distinguish between different levels of synthetic data modification, even with relatively small sample sizes. The behavior of these metrics exhibits consistency across different model sizes. From ESM-2 8M to ESM-2 3B, we see similar patterns in how the metrics respond to increasing sample sizes and perturbation levels.

EMD demonstrates the most pronounced separation between perturbation levels, particularly at smaller sample sizes. However, it also shows the highest variance. In contrast, MMD exhibits the least variance but also the smallest separation between perturbation levels, especially at lower sample sizes. FD strikes a balance between these extremes, offering clear separation with moderate variance.

All metrics show a trend towards stabilization as sample size increases, with diminishing returns beyond $2^{10}$ (1024) samples. This suggests that a sample size of around 1000 sequences may be sufficient for practical applications to obtain reliable distributional distance estimates. The ProtT5 model (Figure 16) exhibits behavior largely consistent with the ESM-2 models, supporting the generalizability of these findings across different protein language model architectures.

Our results suggest that distributional similarity metrics capture the differences in sequence data, even when using relatively small protein language models for embedding. The consistency across model sizes indicates that researchers may be able to use smaller, more computationally efficient models for these analyses.

### 4.4.3 FIDELITY AND DIVERSITY METRICS ANALYSIS

Figure 5 illustrates the responses of Density, IPR Precision, Coverage, and IPR Recall to Cluster Imbalance and Cluster Elimination experiments.

Coverage and IPR Recall exhibit a non-linear response to cluster imbalance, remaining relatively stable until a high degree of imbalance is reached. This behavior suggests these metrics may have reduced sensitivity to subtle distributional shifts in generated samples. Such characteristics could potentially lead to challenges in detecting early stages of mode collapse or minor biases in generative model outputs.

In the cluster elimination scenario, Coverage and IPR Recall demonstrate a more linear decline. While this aligns with expectations, it raises questions about the metrics' ability to differentiate between gradual diversity loss and more abrupt distributional changes. This observation underscores the importance of careful interpretation when using these metrics to assess model performance over time.

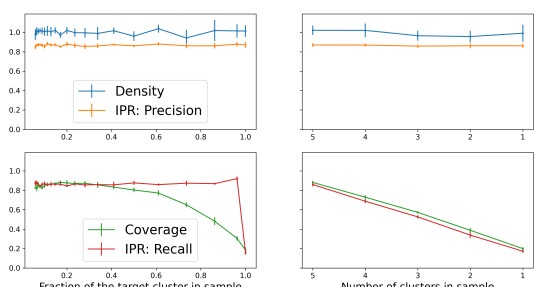

Figure 5: Fidelity and diversity metrics under Cluster Imbalance (left) and Cluster Elimination (right) experiments. Coverage and IPR Recall (bottom) exhibit non-linear responses to imbalance and near-linear decline during elimination.

### 4.4.4 MMD KERNEL PARAMETER ANALYSIS

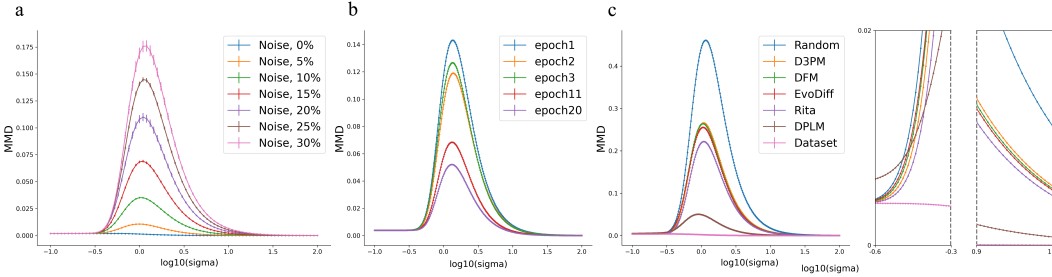

Figure 6: The MMD distance change as a function of the RBF parameter $\sigma$. **(a)** Synthetic data, **(b)** GPT2 training from scratch, **(c)** A series of sequence generative models trained from scratch on SwissProt. Inlays depict switching of model ordering at low $\sigma$ (left) and the ordering at $\sigma = 10$.

The MMD with the Radial Basis Function (RBF) kernel is a widely used metric for assessing distributional similarity, and the choice of the kernel parameter $\sigma$ is crucial, as it can significantly impact the metric's behavior and interpretation. While this issue has been addressed in the context of graph-structured data O'Bray et al. (2022), there is a lack of systematic research on the optimal selection of $\sigma$ in the protein domain. To address this gap, we conduct a comprehensive analysis of the $\sigma$ parameter's impact on MMD in the context of protein sequence evaluation. Our investigation employs three progressively harder experimental settings: **(a)** a controlled scenario with progressive

corruption of high-quality sequences, **(b)** the training progression of a small GPT2 model, and **(c)** the evaluation of various state-of-the-art protein generative models trained from scratch on the same data.

In our controlled corruption experiment (Figures 6a,10), we observe that higher noise levels consistently result in larger MMD values across a wide range of $\sigma$. This behavior demonstrates the metric's sensitivity to data quality. Notably, the relative ordering of noise levels remains stable for most $\sigma$ values, suggesting a degree of robustness in the metric's ability to distinguish between different levels of data corruption.

Figures 6b,11 illustrate the MMD values during the training of a GPT2 model across different epochs. Notably, we observe a critical phenomenon: for $\sigma$ values below approximately 3.2 ($10^{0.50}$), the ordering of epochs 2 and 3 is inverted. This inversion persists until $\sigma$ reaches about 3.5 ($10^{0.55}$), after which the ordering stabilizes and aligns with the results obtained from the Fréchet distance. This observation underscores the sensitivity of MMD to the choice of $\sigma$ and highlights the potential for misinterpretation of model progress if an inappropriate $\sigma$ value is selected.

Figures 6c,12 presents the MMD values for sequences generated by various protein generative models trained from scratch on the same dataset. The results show clear differentiation between models, with random sequences exhibiting the highest MMD values and the trained models achieving lower values, indicating greater similarity to the reference dataset. Crucially, we find that the ordering of models based on MMD is consistent with the Fréchet distance only for $\sigma > 10$. For lower $\sigma$ values, the ordering becomes inconsistent, further emphasizing the importance of appropriate parameter selection.

Based on our analysis, we identify $\sigma = 10$ as the optimal value for the RBF kernel in MMD calculations for protein sequence evaluation. This choice provides a balance between sensitivity to data quality changes while maintaining consistent relative ordering across different experimental settings. Importantly, this value aligns MMD with the Fréchet distance results, providing a convergent perspective on model performance.

### 4.4.5    MMD AS A DIVERSITY MEASURE

Our investigation into the Maximum Mean Discrepancy (MMD) metric reveals its effectiveness as a measure of diversity in generated protein samples. Figure 7 illustrates the response of individual MMD components and the aggregate metric to cluster imbalance and elimination experiments.

In the cluster imbalance scenario, we observe that the RBF(r, r) term, representing the similarity within the reference dataset, remains constant as expected. However, the RBF(q, q) term, denoting the self-similarity of the generated data, increases monotonically with the growing fraction of the target cluster. This indicates that as one cluster becomes more dominant, the generated data becomes more homogeneous. Interestingly, the RBF(q, r) term, which measures the similarity between generated and reference data, shows minimal variation, suggesting that the overall distributional alignment remains relatively stable despite the induced imbalance.

The cluster elimination experiment yields complementary insights. As we progressively remove clusters, the RBF(q, q) term exhibits a marked increase, reflecting the growing self-similarity in the increasingly homogeneous generated data. The RBF(r, r) and RBF(q, r) terms, however, demonstrate relative stability, indicating that the remaining clusters maintain a consistent relationship with the reference distribution.

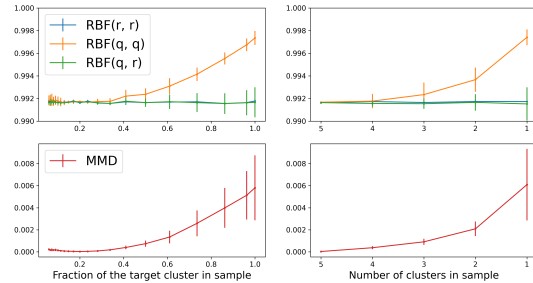

In both scenarios, the aggregate MMD value increases as the perturbations become more pronounced. This trend aligns with our expectation that the metric should capture growing dissimilarity between the generated and reference distributions. The sensitivity of MMD to these controlled perturbations supports its utility in evaluating protein generative models, particularly in detecting mode collapse or overrepresentation of certain protein families.

Figure 7: Analysis of MMD components under cluster perturbation experiments. Cluster imbalance experiment (left). Cluster elimination experiment (right). Individual RBF kernel terms where RBF(r, r) represents similarity within reference data, RBF(q, q) within generated data, and RBF(q, r) between reference and generated data.

These findings underscore the importance of examining individual MMD components alongside the aggregate metric. Such detailed analysis provides deeper insights into the nature of distributional shifts in generated data, potentially guiding more targeted improvements in generative models for proteins.

### 4.5 COMPARATIVE ANALYSIS AND PRACTICAL RECOMMENDATIONS

Our comprehensive evaluation of quality, diversity, and distributional similarity metrics yields several key insights for assessing protein generative models:

**Quality Metrics** For quality assessment, we recommend combining pLDDT and perplexity or scPerplexity alone. While pLDDT and perplexity individually have vulnerabilities (low pLDDT in naturally disordered regions, low perplexity in repetitive sequences), their combination provides a balanced assessment. ScPerplexity mitigates these weaknesses but incurs higher computational costs due to its two-stage procedure.

**Sample Size Considerations** Sample size significantly impacts metric stability. We recommend a minimum of 256 samples for quality metrics, with 512 samples offering more robust estimates, particularly for scPerplexity. Cluster Density calculations stabilize with 500-1000 samples. For distributional similarity metrics, we advise using at least 1024 samples to ensure stable estimates.

**Choice of Models** Our analysis of protein language models (ESM-2 8M, 650M, 3B, and ProtT5) reveals that model choice minimally impacts trends in distributional similarity metrics. This finding suggests that smaller, computationally efficient models often suffice without compromising evaluation accuracy. For structure prediction in quality metrics, ESMFold offers an optimal balance between accuracy and efficiency, making it suitable for large-scale evaluations and real-time assessment during model training.

**Metric Selection** We recommend focusing on MMD with RBF kernel ($\sigma = 10$) and Fréchet Distance (FD) for distributional similarity assessment. These metrics offer a favorable balance between computational efficiency and reliability, especially within the sample size ranges typical for protein evaluations. The choice of metrics should be guided by the specific failure modes researchers aim to detect. Cluster Density at 50% threshold effectively detects mode collapse, while MMD and FD are sensitive to overall distributional changes. Cluster Density at 95% threshold can identify lack of fine-grained diversity within protein families.

**Computational Efficiency** For rapid evaluation during model development, we recommend using perplexity for quality and MMD (with optimized $\sigma$) for distributional similarity. For comprehensive final evaluation, a combination of pLDDT with perplexity, or scPerplexity, Cluster Density, and MMD or Fréchet Distance provides a thorough assessment of model performance. These guidelines aim to balance metric stability, computational efficiency, and comprehensive model evaluation, providing a robust framework for assessing protein generative models across various scenarios and computational constraints.

## 5 CONCLUSION

This study presents a comprehensive analysis of evaluation metrics for protein generative models. We demonstrate that combining quality, diversity, and distributional similarity metrics provides the most robust assessment of generated proteins. Our findings establish practical guidelines for metric selection and sample size determination, balancing accuracy with computational efficiency. These insights contribute to more standardized and meaningful evaluation practices in protein design and generation. As the field advances, continued refinement of these evaluation methods will be crucial for guiding the development of increasingly sophisticated protein generative models.

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

## A  METRICS

The evaluation of protein generative models encompasses three primary aspects: quality, diversity, and distributional similarity. This section outlines the theoretical foundations of commonly used metrics in each category.

### A.1  QUALITY METRICS

Quality metrics assess the characteristics of individual generated proteins. We examine four widely used metrics: pLDDT, perplexity, pseudoperplexity, and scPerplexity. These metrics evaluate protein quality from perspectives of structural stability and sequence plausibility.

**pLDDT.**  Quality metrics assess individual generated proteins, focusing on structural stability and sequence plausibility. Four metrics are frequently employed in the literature: pLDDT, perplexity, pseudoperplexity, and scPerplexity. The predicted Local Distance Difference Test (pLDDT) evaluates structural quality Jumper et al. (2021). For a protein with $N$ residues, pLDDT is defined as:

$$\text{pLDDT} = \frac{1}{N} \sum_{i=1}^{N} \text{pLDDT}_i \tag{1}$$

where $\text{pLDDT}_i$ is the score for the $i$-th residue. Perplexity (ppl), adapted from language modeling, assesses sequence quality Madani et al. (2023). For a protein sequence $S = (s_1, s_2, ..., s_N)$, perplexity is calculated as:

$$\text{ppl}(S) = \exp\left(-\frac{1}{N} \sum_{i=1}^{N} \log p(s_i|s_1, ..., s_{i-1})\right) \tag{2}$$

where $p(s_i|s_1, ..., s_{i-1})$ is the probability of the $i$-th amino acid given the preceding sequence. Pseudoperplexity (pppl) extends perplexity to masked language models Salazar et al. (2019). For a sequence $S$ and model parameters $\Theta$, pppl is defined as:

$$\text{pppl}(S) = \exp\left(-\frac{1}{N} \sum_{i=1}^{N} \log p(s_i|S_{\backslash i}, \Theta)\right) \tag{3}$$

where $S_{\backslash i}$ is the sequence with the $i$-th residue masked. Self-consistency perplexity (scPerplexity) incorporates the sequence-structure relationship Alamdari et al. (2023):

$$\text{scPerplexity}(S) = -\log p(S|G(F(S))) \tag{4}$$

where $F$ is a folding model and $G$ is an inverse folding model.

### A.2  DIVERSITY METRICS

Cluster Density (CD) is used to assess the diversity of generated sequences:

$$CD = \frac{\text{Number of Clusters}}{\text{Total Number of Sequences}} \tag{5}$$

Clustering is typically performed using MMseqs2 Steinegger & Söding (2017) at various similarity thresholds.

### A.3  DISTRIBUTIONAL SIMILARITY METRICS

Distributional similarity metrics compare generated and real protein distributions. Both direct measurement approaches and compound metrics are used in the field. The Improved Precision and Recall (IPR) method Kynkäänniemi et al. (2019) directly assesses fidelity and diversity:

$$\text{precision}(\Phi_r, \Phi_g) = \frac{1}{|\Phi_g|} \sum_{\phi_g \in \Phi_g} f(\phi_g, \Phi_r), \quad \text{recall}(\Phi_r, \Phi_g) = \frac{1}{|\Phi_r|} \sum_{\phi_r \in \Phi_r} f(\phi_r, \Phi_g) \tag{6}$$

where $f(\phi, \Phi)$ determines whether a sample $\phi$ falls within the manifold defined by set $\Phi$. The Fréchet distance quantifies dissimilarity between multivariate Gaussian distributions:

$$d(X_1, X_2)^2 = ||\mu_1 - \mu_2||^2 + \text{Tr}(\Sigma_1 + \Sigma_2 - 2\sqrt{\Sigma_1 \Sigma_2}) \tag{7}$$

for samples $X_1 \sim \mathcal{N}(\mu_1, \Sigma_1)$ and $X_2 \sim \mathcal{N}(\mu_2, \Sigma_2)$. Maximum Mean Discrepancy (MMD) Gretton et al. (2012) measures distributional distance in a reproducing kernel Hilbert space:

$$MMD_k^2(X, Y) = \frac{1}{n^2} \sum_{i=1}^{n} \sum_{j=1}^{n} (k(xi, xj) + k(yi, yj) - 2k(xi, yj)) \tag{8}$$

for samples $X = x_1, ..., x_n$, $Y = y_1, ..., y_n$, and kernel $k$. The Earth Mover's Distance (EMD) quantifies the minimum cost of transforming one distribution into another. These metrics form the basis for evaluating protein generative models in terms of quality, diversity, and distributional similarity. Their practical application and limitations are explored in subsequent sections.

# B   ADDITIONAL EXPERIMENTS

Figure 8: Pseudoperplexity as a function of ESM-2 model size.

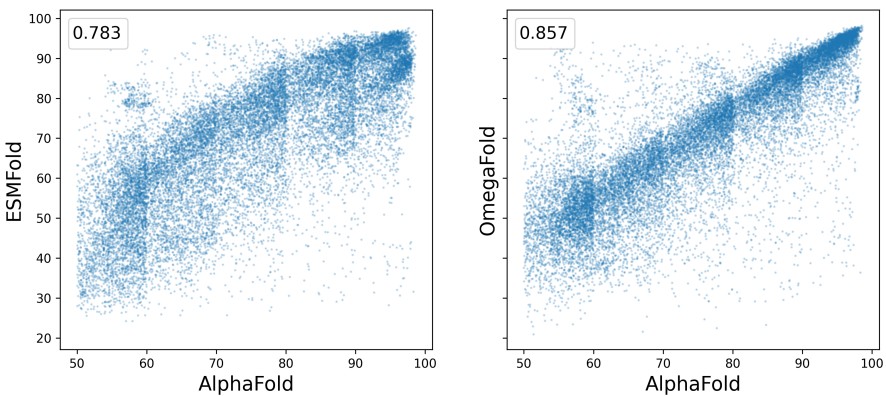

Figure 9: The correlation between pLDDT values produced by AlphaFold, ESMFold, and OmegaFold models.

## B.1   DEPENDENCE OF EMBEDDING MODEL AND SAMPLE SIZE ON DISTREIBUTION SIMILARITY METRICS.

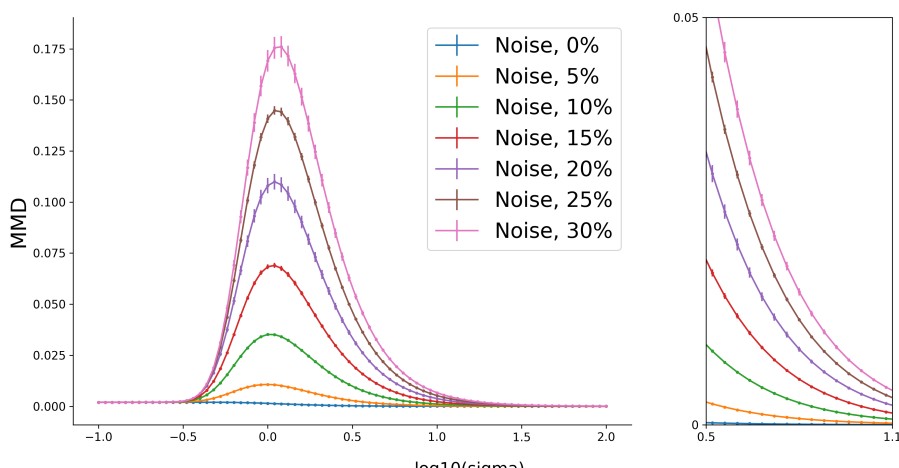

Figure 10: The MMD distance change between the original dataset and its progressively corrupted version as a function of the RBF kernel parameter $\sigma$.

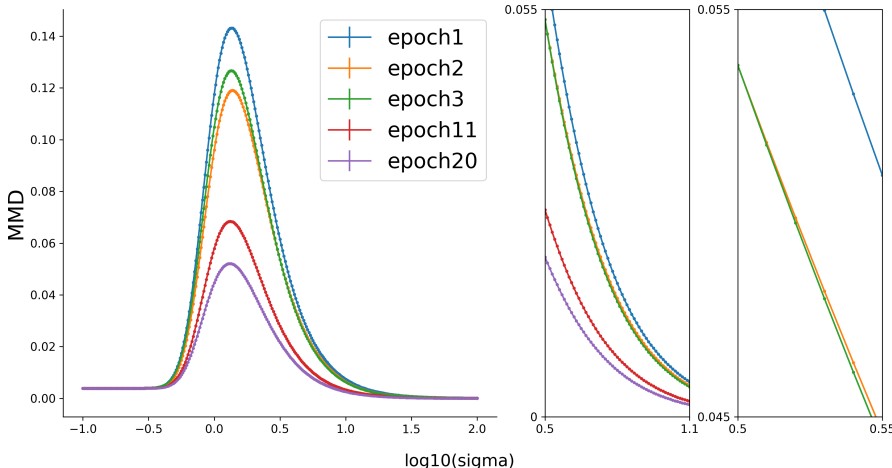

Figure 11: The MMD distance of the samples generated during the model training to the training data as a function of the RBF kernel parameter $\sigma$.

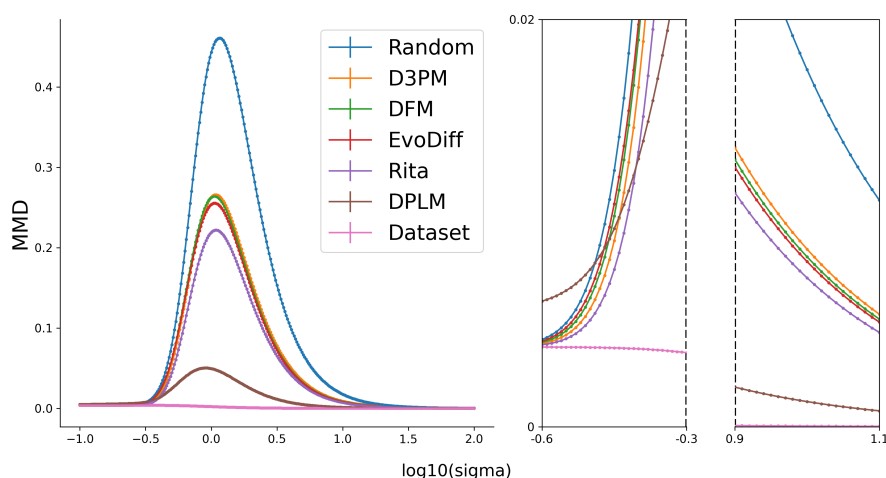

Figure 12: The MMD distance between the data produced by a series of generative models to the training data as a function of the RBF kernel parameter $\sigma$.

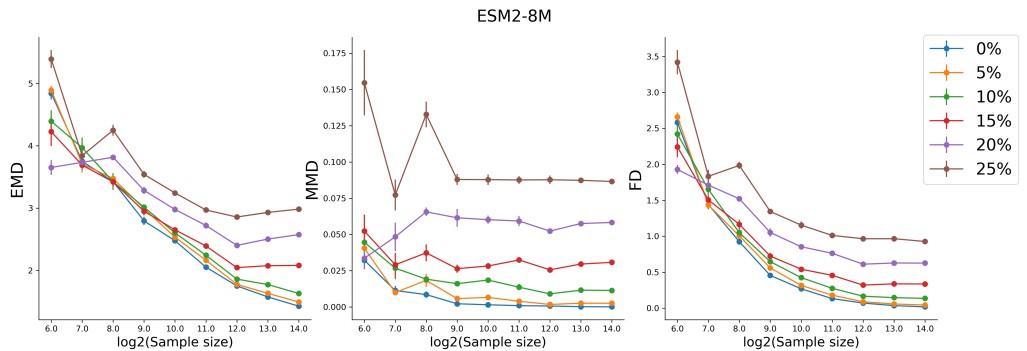

Figure 13: The distance metrics on corrupted sequences for ESM-2 8M model.

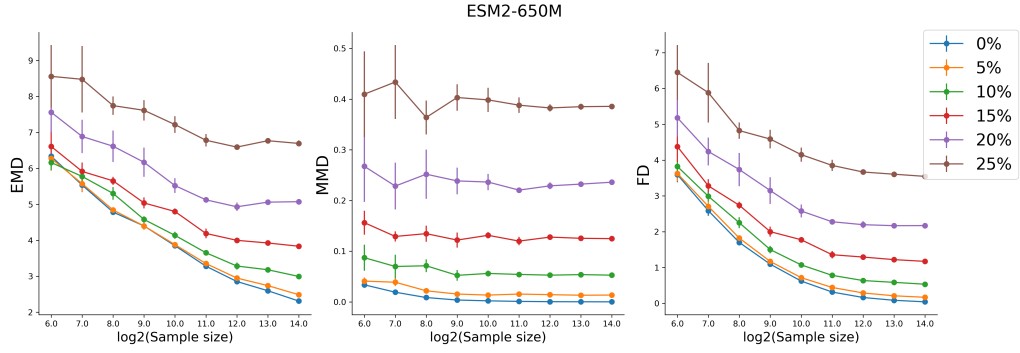

Figure 14: The distance metrics on corrupted sequences for ESM2-650M.

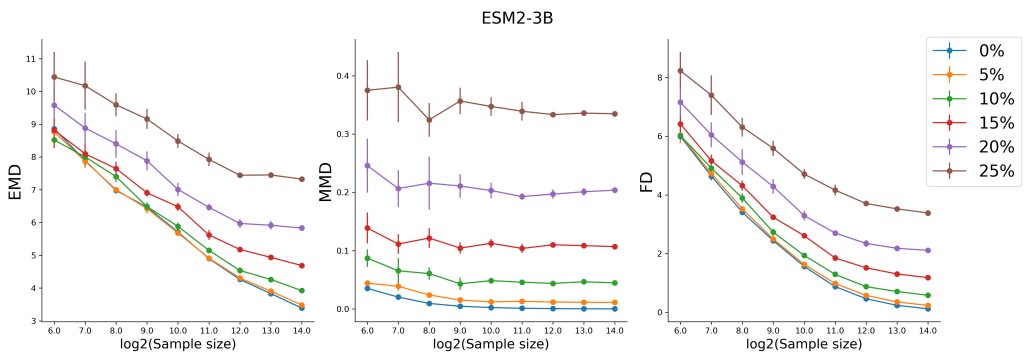

Figure 15: The distance metrics on corrupted sequences for ESM2-3B.

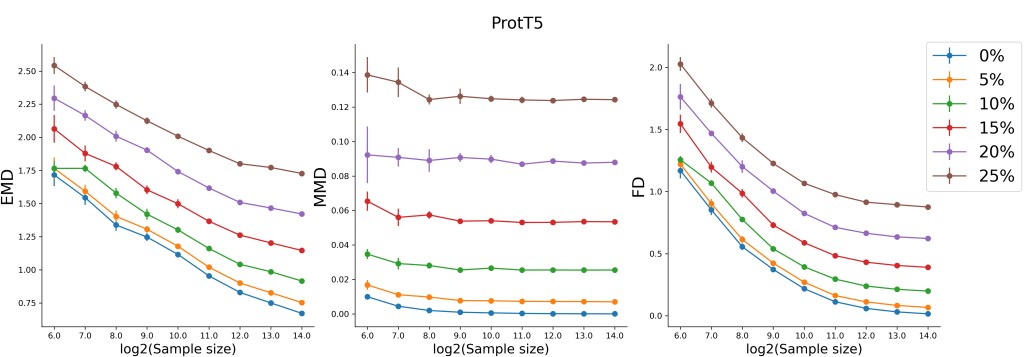

Figure 16: The distance metrics on corrupted sequences for ProtT5.

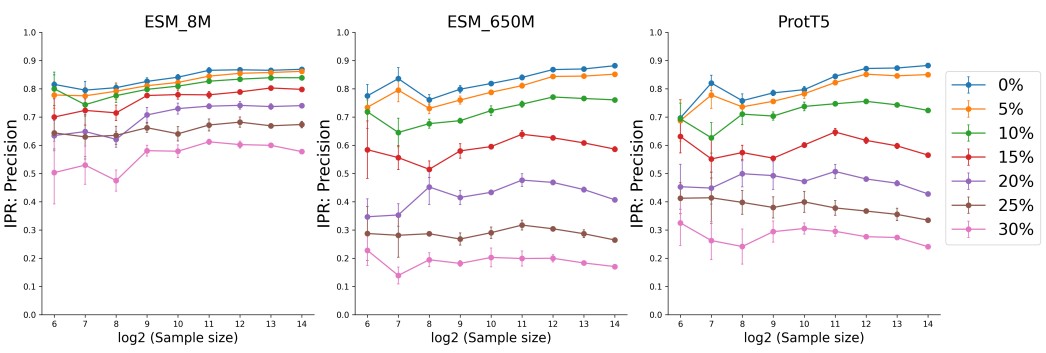

Figure 17: 'IPR Precision on corrupted data using ESM 8M, ESM 650M and ProtT5 embeddings.'

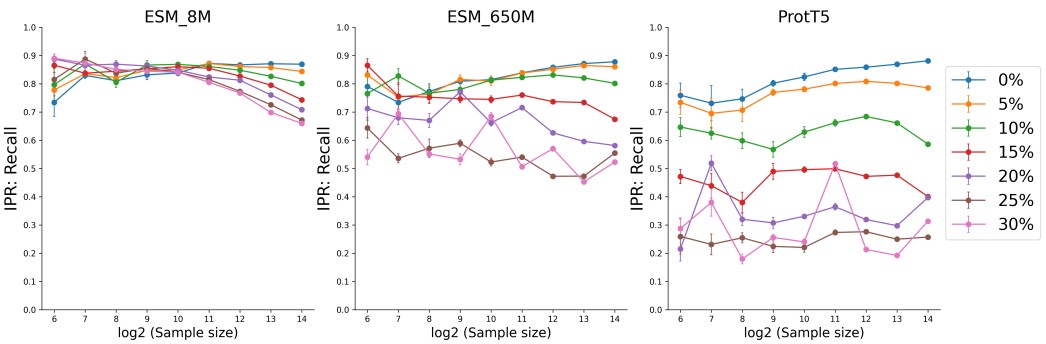

Figure 18: 'IPR Recall on corrupted data using ESM 8M, ESM 650M and ProtT5 embeddings.'

