# OpenReview forum: "Towards Robust Evaluation of Protein Generative Models: A Systematic Analysis of Metrics"
_ICLR.cc/2025/Conference — Submitted to ICLR 2025_

### Official Review · Reviewer_Rn7f · 2024-10-16

**Soundness:** 1
**Presentation:** 2
**Contribution:** 1
**Rating:** 3
**Confidence:** 5

**Summary:**

This work evaluates various quality, diversity and distributional similarity metrics for their ability to co-vary with random synthetic perturbations on protein amino acid sequences. The authors also evaluate differences in plddt scores for forward folded noised sequences.
The stated aim is to provide a systematic overview of how those metrics change with sequence randomness (noise), number of protein samples and model size to advance the evaluation of protein generative models. However, I think this works falls short of this aim. The proposed metrics are omitting the non-bijective nature of the protein structure-sequence relationship, the authors do not compare with well-established quality metrics in the field of generative protein design (e.g. self-consistency folding as a quality metric for structural fidelity, or edit-distance as a function of distributional similarity). The authors only present results on synthetically perturbed sequences, where residues are mutated with equal probability to assess perplexity and diversity. This is very different from the case of generative modeling, where diverse sequences are generated (non-randomly!) via auto-regressive sampling, any-order sampling or temperature sampling. I would recommend to generate sequences with these models, and synthetically perturbed sequences with BLOSUM or PAN transition matrices.
I don't think that machine-learning motivated metrics, such as perplexity, or earth mover's distance are practically useful for the field of generative protein design. Useful metrics should capture if the model generates protein sequences or structures, that fold, are stable, exhibit a specific function.
I have several concerns about the methodology, biological soundness and presentation of this work as I will outline concretely below.

**Strengths:**

Originality: A systematic analysis of metric variability with protein sequence diversity is a good idea and I would recommend the authors to build on it, but incorporate several improvements: Instead of uniform probabilities for mutations it might be more meaningful to use PAM or BLOSUM matrices. These are the transition probabilities from one amino acid residue to another one (based on similar hydrophobicity, charge, polarity, size etc.).
Significance: The authors correctly emphasize that there is no gold-standard in the field of generative protein models on what constitutes a "good" protein. The topic is worth being addressed, although I don't think that this work provides a significant contribution.

**Weaknesses:**

Background section:
1. The motivation of diversity in the absence of training data is confusing. The authors should discuss that the structure-sequence relationship is no bijective and a one-to-many mapping problem. There are very many sequences that fold into the same or similar structures. The true diversity of this solution space is not known, given the small size of structural data in the PDB. This diversity is likely a complex function of protein size (there are very many diverse sequences that all fold into the same alpha helix peptide), packing (internal residues less diverse, versus external residues etc.
2. The authors mention structural stability as a measure of a "good" protein, but do not evaluate this property in this work, this is confusing.
3. I find the mathematical notations (especially under "self-consistency" overly complicated (given they are not being used) anywhere else.
Section 3, Metrics:
1. Fidelity metrics: The fidelity metrics are not addressing structural fidelity in terms of structural similarity (e.g. TM-score or RMSE) in the case of forward folding. Or self-consistency TM in the case of inverse folding.
2. In general I would recommend the authors to split metrics for different generative model types and approaches, e.g. sequence-based (e.g. LLMs), inverse folding (structure-to-sequence)
3. In section 2.3. the authors state that metrics should be interpretable. I don't find perplexity, or pseudo-perplexity very interpretable. plddt is interpretable. I would recommend adopting metrics like edit distance or structure consistency (e.g. TMscore). I think reporting perplexity in a protein LLM is still valuable, but it's not particularly novel or insightful. I am not sure if self-consistency perplexity: -logp(S|G(F(S)) makes sense given that this protein inverse folding (G) is a one-to-many problem with an unknown and variable number of diverse solutions. And as the authors state -- the folding and inverse folding model bias might further complicate this metric.
4.  Section 3.2: The diversity defintion of cluster density at 50% and 95% is interesting, but shoudl be compared to more commonly adopted diversity metrics in the field, such as edit distance and pairwise distances.

Section 4: Experiments:
1. I like the idea of a systematic perturbation of amio acid sequences, but random noise (uniform transition probabilities) is unrealistic. I would recommend using BLOSOM or PAN matrices. Additionally to the synthetic perturbations I am missing an actual application to generative models. I would recommend using different inverse folding models, e.g. ESMIF or ProteinMPNN and generating diverse sequences with random decoding orders and temperature sampling. Currently the authors perturb the sequence in a random way which likely turns them easily into garbage (ie they would never exist in nature and fold).
2. The random perturbations do not create meaningful biological diversity in the sequences and simply degrade their quality. As such Figures 2 and 4 are stating obvious trends: The more noise, the worse the quality/fidelity metrics.

Presentation:
1. Please review citation guidelines, current citation style is reader unfriendly.
2. Please mark supplementary figures as such (e.g. Figure 9).
3. Figure 8 is missing

**Questions:**

Similar to above weaknesses:
1. How do those metrics behave for meaningful diverse sequences, that were note generated with random noising?
2. Are the randomly noised sequences foldable? Have you tried to calculate the TM-score between the original sequence and the forward folded structure of a 30% noised sequence?

---

### Official Review · Reviewer_Hg9Y · 2024-10-27

**Soundness:** 1
**Presentation:** 1
**Contribution:** 1
**Rating:** 1
**Confidence:** 5

**Summary:**

The present work attempts to provide insight into various metrics of generative models over protein sequences. They evaluate several metrics used in prior works such as predicted local distance difference test, perplexity, pseudo perplexity, self-consistency perplexity, cluster density, and multiple techniques for distributional similarity metrics. On a curated dataset, they measure robustness to random perturbations, sensitivity to sample size, and use of different protein language models to compute the metrics. Some recommendations are provided at the end of which models to use and sample size for robust evaluation.

**Strengths:**

* Evaluation of robustness to several protein sequence generation metrics.

**Weaknesses:**

The present work contains no technical novelty or new results. Therefore the analysis and presentation needs to be of high quality. Unfortunately the presentation quality is low and the insights are novel enough for acceptance to ICLR.

* First, the work claims to evaluate protein generative models but proceeds to ignore or miss protein structure generative models such as RFdiffusion [1], Chroma [2]. The work only attempts to evaluate protein sequences without consideration for generated structures. Considering the popularity and success of [1, 2], this is a major omission.

* There are **no benchmarks** of generative models in this work. The experiments are conducted on artificial perturbations of known sequences and on a curated set of sequences from 5 protein families. The insights in this work cannot be believed and are of little use unless the metrics are rigorously evaluated on state-of-the-art protein generative models.

* Metrics are only useful if they correspond to success in downstream applications. The metrics used in [1, 2] are accepted because they are known to correlate (albeit weakly) with experimental success [3]. None of the metrics utilized in this work are associated with success in downstream applications. Indeed we care about how well the samples capture distributions but they are auxiliary metrics and are not the primary metrics in high impact protein generative model publications.

* The noise perturbations are artificial. How do we know if randomly mutating 5-30% of the sequence is a failure mode or common occurrence in existing protein generative models?

* Novelty is mentioned as a important consideration but no novelty metrics are presented or discussed.

* Only using 5 protein families is far too small of an evaluation set. Line 234 states the experiments are done on "real-world generated data" but what is actually being generated here?

* Section 4.3 on diversity metric analysis is weak. The trend in Figure 3 is the expected behavior of the 50% and 95% sequence similarity threshold. There is no new insight here.

* I'm not sure what new insight is provided from the noise. Figures 2 and 3 show more noise leads to all the metrics becoming worse. This is expected but there is no indication of how this transfers to commonly used protein generative models. Do protein generative models exhibit such behavior?

* Section 4.4 is also weak on insights. The graphs are expected by changing the noise and RBG kernel width. It would seem to me that different downstream applications would call for different parameters and robustness. Instead, the claims here are too general and unclear how useful they are for specific downstream applications such as binder design.

* I would have liked to see a ranking of protein sequence generative models such as ESM2, ProGen, T5 with the metrics provided.

Overall I do not believe this work provides a careful and rigorous study of evaluating protein generative models. I recommend the authors to rethink the experiments and hypotheses they wish to test.

[1] https://www.nature.com/articles/s41586-023-06415-8
[2] https://www.nature.com/articles/s41586-023-06728-8
[3] https://www.nature.com/articles/s41467-023-38328-5

**Questions:**

Many of my questions are embedded in the weaknesses. Some more minor questions.

* Line 37. What is the missing reference "?"

* Line 88. Self-consistency is mentioned but this equation is never used. Why is this given and where is it actually used?

* Line 158. The equation $-\log p(S|G(F(S))$ is confusing. If $G(F(S))$ is the inverse folding prediction then what does it mean to conditioned $p(S| \cdot)$ on this?

* Line 230. What protein generation tasks are considered?

* Line 433. How are "state-of-the-art protein generative models" re-trained?

---

### Official Review · Reviewer_xLmN · 2024-10-28

**Soundness:** 2
**Presentation:** 3
**Contribution:** 2
**Rating:** 3
**Confidence:** 4

**Summary:**

The paper analyses several metrics for protein generation on synthetic datasets with controlled properties in order to see their strengths, limitations, and practical applicability. The paper highlights that some metrics are dependant on sample size and that computationally efficient metrics can be just as effective.

**Strengths:**

- The paper's background section and motivation is extremely strong. The need for reliable evaluation metrics for protein generation is convincing and some of the metrics used in the literature are clearly outlined.

- The controlled experiments are well thought out and provide some useful information about the quality of the metrics.

- The authors perform a rigorous set of experiments and the provided practical recommendations could be useful to the community.

**Weaknesses:**

- The main weakness comes from evaluating the metrics in such a controlled and synthetic setting. The quality metrics are evaluated on proteins which the models (such as ESMFold) are trained on. In this case, introducing more noise is shown to cause the metrics to get worse. In practice though, we generate unseen proteins and it is not clear whether these metrics generalize to proteins they are not trained on. Additionally, it is not clear from the paper whether these metrics correlate with anything experimentally.  Therefore, the evaluation of these metrics in the given scenario doesn’t seem to offer much practical insight on the usefulness of these metrics.

- The authors compare different metrics and explain that there should be a tradeoff with computational efficiency. However, it is not clear how the methods actually differ in this regard. You mention a few times that scPerplexity is expensive to calculate as it involves two models but there is no figure or timing comparison given. How much slower is it and is it impractical? You also say that your proposed metrics allow for rapid evaluation. Again, how long are these proposed metrics taking and what does the term “rapid” quantitatively mean? Although computational efficiency is mentioned a lot throughout the work, and seems to be important for selecting metrics, I have no indication from the paper on how these methods actually differ in this regard and why I should use a method over another practically.  To improve this, the authors could include a table or figure comparing the runtime of each metric on a standardized dataset, perhaps across different sample sizes.

**Questions:**

- In line 307 you say that ScPerplexity has the highest sensitivity to sample size. Firstly, this isn’t clear from the plots as it doesn’t seem to change any more than plDDT. Additionally, why does sample size matter if the ordering with respect to noise is always correct? In practice, we can fix the sample size and correctly rank different generative models.

- You say one of the important aspects of a metric is its interpretability but this isn’t considered later when evaluating the metrics. Are these metrics interpretable and are there differences in interpretability between them?

- You say that a good generated protein should be structurally stable. Are any of the metrics actually capturing this?

minor comments

- Line 37 missing reference

- Quotation marks are always the wrong way up when used. For example, line 46.

- Some references are missing their journal. For example, “Generating novel, designable, and diverse protein structures
by equivariantly diffusing oriented residue clouds” was at ICML 2023.

---

### Official Review · Reviewer_YRsS · 2024-10-29

**Soundness:** 2
**Presentation:** 2
**Contribution:** 3
**Rating:** 5
**Confidence:** 4

**Summary:**

This paper studies several common evaluation metrics for protein sequence generative models covering quality, diversity and distributional similarity of samples.
The authors present controlled experiments and derive guidelines for a robust assessment of the performance of protein generative models.

**Strengths:**

### Importance

Unifying benchmarking attempts for protein generative models is an extremely important open challenge.
Studying various common evaluation metrics systematically and in a controlled setup is impactful because it can inform future developments of new methods and allow researchers to benchmark their models in a more convincing way.
The problem is motivated nicely and grounded in related works.

### Breadth

The paper addresses three dimensions of generative model evaluation: **quality**, **diversity**, and **distributional similarity**.
It furthermore identifies at least two axes along which evaluation metrics should be assessed: **robustness vs sensitivity** and **reliability vs computational efficiency**.
Together these cover most practically relevant aspects of model evaluation in this space.

**Weaknesses:**

### Clarity

The presented topic is very complex and the authors' attempt to illuminate the design space for these metrics from various angles is commendable.
However, the clarity of the presentation of their results can be improved.
The paper introduces a lot of metrics and desirable properties thereof but the arguments are sometimes difficult to follow in the current state.
It could be useful to restructure the experimental results section so that each subsection (quality, diversity and distribution similarity) systematically analyses different available metrics regarding their (1) robustness-sensitivity trade-off, and (2) reliability-efficiency trade-off.
I would define a clear, quantitative criterion for each and follow an identical structure in each subsection (quality, diversity and distribution similarity).
The current discussion sometimes mixes empirically supported findings with intuition-derived arguments.

In the background section, it is confusing that most of the time the paper discusses three key axes of model performance: quality, diversity and distribution similarity,
but in Section 2.2 it talks about an alternative set of objectives: fidelity, diversity, novelty.
Similarly, the paper introduces "Interpretability" in Section 2.3 but does not discuss this aspect in the Results section.
I would recommend to be more consistent throughout the paper (both in terms of wording and semantics).

Furthermore, the paper should define the scope of the work clearly. It only covers generative models for amino acids _sequences_ as opposed to backbone _structures_.
The discussion about self-consistency in Section 2.1 seems unnecessarily detailed given the concept is only used once later on (scPerplexity metric).
When I arrived at this point in the manuscript I was under the impression that the paper discusses both sequence and structure generative models because self-consistency is primarily used in the evaluation of _structure_ design methods (e.g. [1]).



### Analysis of diversity metrics

The analysis of diversity metrics (Section 4.3) is extremely short, and it is unclear whether the presented data in Figure 3 provides information about the _sensitivity_ or _robustness_ of the Cluster Density metric.
The absence of a comparison with alternative approaches additionally makes it hard to interpret the results.


### Support every claim with empirical data

A systematic evaluation of metrics should always provide empirical evidence to back up the presented conclusions.
Here, this is missing in some cases. For instance,
- Looking at Figure 9 I would argue there are still notable differences between AlphaFold2 and ESMFold. Rather than just assessing their correlation, it would be useful to understand how sensitive and robust each method is to sample quality differences.
- The paper states that simple diversity metrics lack discriminative power but it does not discuss any examples in the analysis in Section 4.3.
- The paper also mentions intrinsically disordered regions as a potential stumbling block for the pLDDT metric. While this assumption is reasonable, it is still possible that pLDDT has better discriminative power than alternative metrics in those cases, but only empirical data can provide an answer to this question.
- Finally, statements about computational efficiency are never quantified. Providing concrete run times would be an important piece of information that allows readers to get an idea about the reliability-efficiency trade-off.



### References

[1] Yim, Jason, et al. "SE (3) diffusion model with application to protein backbone generation." arXiv preprint arXiv:2302.02277 (2023).

**Questions:**

- Would it be possible to discuss the sensitivity-robustness trade-off more systematically & quantitatively? For instance, does it make sense to interpret the cluster elimination experiment as a strong perturbation (that a sensitive distribution similarity metric should detect) and intra-cluster diversity reduction as a weak perturbation (that distribution similarity metrics should be more or less robust to)?

- Why is the CD diversity metric not compared to simpler alternatives like average pairwise distances between generated sequences?

- I would like to see some reference data for run times of different metrics to support statements about the reliability-efficiency trade-off.

- Section 4.4.3 should also discuss the _Density_ and _IPR Precision_ results.

- The conclusion states "We demonstrate that combining quality, diversity, and distributional similarity metrics provides the most robust assessment of generated proteins". As far as I can tell all experiments evaluate metrics in isolation and therefore do not really support this statement. Could you please elaborate a bit more?

- Figure 8 is missing.

### Minor comments

- line 37: missing/broken reference
- line 43: reference seems to be incorrectly formatted
- quotation marks should be corrected in some places (e.g. lines 73 and 81)
- the norm in the equation in line 89 is not specified, maybe a more general notation for a distance function should be used here
- line 103: I am not sure if I agree with the definition of _diversity_ using memorization of the training data. Samples from the training set can still be diverse. Doesn't this definition apply to _novelty_?
- In many places, it would be preferable to change the formatting of citations (use `\citep` instead of `\citet`).
- line 157: why did the notation change? Before, small letters were used to denote the folding function and inverse folding function, respectively.
- line 214: indices are not correctly formatted
- Figures 2 - 5: error bars should be defined in the figure legends
- Figure 4 should be referenced in the main text.
- Section 4.2.1: how many data points were used to calculate the correlation values? Is the raw data shown somewhere?

---

### Meta-Review · Area_Chair_2sKs · 2024-12-27

**Metareview:**

This paper examines evaluation metrics for protein generative models, looking at quality, diversity, and distributional similarity criteria. Proteins are essential to life, and AI has shown great promise in biology with the advent of AlphaFold and other state-of-the-art protein models (both at the sequence and structure levels). Several metrics are used in the literature to evaluate these protein models. Analyzing these metrics is therefore important. While the reviewers agree on the importance of this topic, they judged the paper, in its present state, misses several key elements. Indeed, it failed to include some metrics in the evaluation set, it looked only at synthetic datasets and used perturbation schemes for protein sequences such as random noise that might be unrealistic. Reviewers were also unhappy with the lack of clarity of the presentation.

**Additional Comments On Reviewer Discussion:**

The authors did not provide any rebuttal, comment, or response to the reviews. The reviewers raised valid points that were not addressed.

---

### Decision · Program_Chairs · 2025-01-22

Reject